# Graphene in Solid-State Batteries: An Overview

**DOI:** 10.3390/nano12132310

**Published:** 2022-07-05

**Authors:** Syed Atif Pervez, Milad Madinehei, Nima Moghimian

**Affiliations:** NanoXplore Inc., 4500 Thimens Blvd, Saint-Laurent, QC H4R 2P2, Canada; miladmadinehei@gmail.com (M.M.); nima.moghimian@nanoxplore.ca (N.M.)

**Keywords:** solid-state battery, solid electrolyte, graphene, interface, Li dendrites, energy storage

## Abstract

Solid-state batteries (SSBs) have emerged as a potential alternative to conventional Li-ion batteries (LIBs) since they are safer and offer higher energy density. Despite the hype, SSBs are yet to surpass their liquid counterparts in terms of electrochemical performance. This is mainly due to challenges at both the materials and cell integration levels. Various strategies have been devised to address the issue of SSBs. In this review, we have explored the role of graphene-based materials (GBM) in enhancing the electrochemical performance of SSBs. We have covered each individual component of an SSB (electrolyte, cathode, anode, and interface) and highlighted the approaches using GBMs to achieve stable and better performance. The recent literature shows that GBMs impart stability to SSBs by improving Li^+^ ion kinetics in the electrodes, electrolyte and at the interfaces. Furthermore, they improve the mechanical and thermal properties of the polymer and ceramic solid-state electrolytes (SSEs). Overall, the enhancements endowed by GBMs will address the challenges that are stunting the proliferation of SSBs.

## 1. Introduction

A Li metal-based SSB is one of the leading contenders to make electric vehicles mainstream [1,2]. In an SSB, the organic liquid electrolyte is replaced with a non-flammable SSE. The use of SSE helps in integrating Li metal as an anode, which is considered the holy grail of all the anode materials due to its high theoretical capacity (3860 mAh g^−1^) and the lowest redox potential (−3.04 V vs. SHE) [3]. Replacing graphite anode with Li and using a very thin (few tens of microns) SSE, between 70 and 40% increase in volumetric and gravimetric energy density on cell level is possible. Previously, Li metal anode has been used with organic liquid electrolytes; however, its practical application has been severely hindered by problems such as poor cycle life and safety concerns that originated from its high and undesired reactivity with organic liquid electrolytes and uneven deposition behavior (Li dendrites) [4]. Li dendrites are generally mossy- or needle-like structures that initiate from the Li metal surface, propagate through the electrolyte during successive cycles and carry the risk of short-circuiting the battery by puncturing the polymer separator [5]. Based on the Monroe and Newman model, the minimum shear modulus should be at least double that of Li metal to ensure mechanical blockage of Li dendrites [6]. In this regard, most of the SSEs, in particular, ceramic SSEs, are above this mark.

SSEs are mainly classified into the following three categories: ceramic, polymer, and hybrid solid polymer electrolytes. Ceramic electrolytes have generally two orders of magnitude higher room temperature ionic conductivities (10^−3^–10^−2^ S cm^−1^) compared to polymer electrolytes. They also typically exhibit high transference numbers (close to unity) and shear modulus (of the order of GPa). However, they suffer from poor interfacial ionic kinetics due to the nature of solid-solid electrode|electrolyte interfaces [5,7]. Promising classes of ceramic electrolytes are sulfide/thiophosphate (LGPS, LiSiPS, LiPS, Li_2_S–P_2_S_5_ etc.) and oxides (LLZO, NASICON, LISICONs etc.) [8,9,10] Ionic conductivities of oxides are generally an order of magnitude lower than sulfides, but they offer better electrochemical stability with Li metal [1]. Solid Polymer electrolytes (SPEs), on the other hand, are much easier to process and are fabricated generally by solution casting, making them compatible with commercial roll-to-roll battery manufacturing processes. Moreover, due to their higher elasticity and plasticity, stable interfaces can be realized that can buffer the volume changes occurring on a microscale during cell operation [11]. Hybrid or composite electrolytes combine the advantages of both the ceramic and polymer systems to achieve improved performance [12]. Particles that are either ionically conducting ((Li_7_La_3_Zr_2_O_12_; LLZO), Li_1+x_Al_x_Ti_2x-x_(PO4)_3_; LATP), (Li_2_S-P_2_S_5_; LPS), etc.) or non-conducting (Al_2_O_3_, TiO_2_, SiO_2_, MgO, etc.) [13,14,15,16] are added as fillers to the polymer matrix to enhance their mechanical strength and ionic conductivities. Depending on the polymer/inorganic ratio, the terminologies such as ceramic-in-polymer and polymer-in-ceramic are often used. Other techniques to tune the properties of hybrid SPEs include copolymerization [17], crosslinking [18], interpenetration and blending [19]. Apart from tuning the electrolyte properties, its integration with the Li anode and various cathodes is also a challenging task. This is mainly due to the nature of the solid–solid interface where intimate contact between electrode and electrolyte is not easy to achieve [20]. Interface engineering and design are therefore crucial in ensuring the stable performance of SSBs as will be discussed in the following sections.

Graphene has been deemed a wonder material since its discovery. It is a single atomic layer of graphite consisting of sp2-hybridized carbon atoms packed into a honeycomb lattice [21]. In recent years, graphene has grabbed great technological interest due to its unique traits [22]. It exhibits amazing electronic properties such as extremely high charge carriers (electrons and holes) mobility = 230,000 cm^2^ V^−1^ s^−1^ at room temperature, excellent thermal conductivity = 5000 W m^−1^ K^−1^, and mechanical stability (Young’s modulus of 1 Tpa). Due to its monolayer structure, graphene has a very high specific surface area of 2630 m^2^ g^−1^. This is much larger than that reported to date for carbon black (typically smaller than 900 m^2^ g^−1^) or for carbon nanotubes (CNTs), ranging from ≈ 100 to 1000 m^2^ g^−1^ and is similar to activated carbon. The graphene sheet is a semi-metal (or a zero-gap semiconductor) because its conduction and valence bands meet at the Dirac points [22]. Graphene can also be modified to generate a band gap (in the range from 0 to 0.25 eV) that can lead to application in the semiconductor industry for developing devices such as transistors. Further, graphene or GBMs exhibit novel electrochemical properties such as low charge transfer, wide potential window, excellent electrochemical activity, and fast electron transfer rate [23,24,25,26]. Such properties make GBM, including graphene oxide (GO), reduced graphene oxide (r-GO), few-layer graphene (FLG), and graphene nanoplatelets (GNP), highly suitable for solid-state battery applications.

Herein, we provide a comprehensive overview of the recent reports published on the use of GBMs in SSBs. The sections are arranged to cover the following three main components of an SSB: electrolyte, electrodes and electrode|electrolyte interfaces. Based on the available literature as summarized in this article, GBMs are important materials to substantially enhance the properties of electrodes (Li, Sulfur, transition metal oxides) and electrolytes (polymer, ceramic, and hybrid). Furthermore, GBMs also play a vital role in improving the interfacial kinetics of Li^+^ ions at the solid–solid interfaces. Figure 1 depicts schematic of a solid-state battery and lists GBMs improvements relevant to electrodes, electrolyte and the interfaces.

## 2. Synthesis Methods of Graphene-Based Materials

Since the successful exfoliation of isolated graphene in 2004, one of the major challenges has been finding a fabrication method that can not only produce high-quality graphene but also on a large scale. There are the following two major pathways of graphene synthesis: ‘Bottom-up’ and ‘Top-down’ methods [25,27]. The first method makes wide graphene films grow on top of the metallic foils, while the ‘top-down’ method is carried out through exfoliation by mechanical or chemical pathways to separate the carbon layer from the structure of graphite or graphite oxide. Depending on the method, different types of graphene can be prepared in terms of scale, exfoliation degree, purity, and structural defects [28,29].

In the following, we introduce the graphene derivatives for battery applications and their most common preparation methods.

### 2.1. Chemical Vapor Deposition

The CVD of graphene films on metal substrates such as copper and nickel has demonstrated great potential to supply the increasing demand for next-generation electronics [30]. These graphene films ideally consist of either a single layer or a few layers of pure graphene. Generally, the electrical conductivity of graphene synthesized by CVD is higher than that synthesized by the chemical method, and no reduction process is needed. There are numerous types of CVD techniques available, such as plasma-assisted CVD, thermal CVD, hot or cold wall CVD, etc. These methods are carried out using highly volatile carbon sources (e.g., methane), under a harsh environment, with the use of inert carrier gases. Although many advancements have been made, the processes listed above are not commercially viable yet due to high costs and complexities.

For the CVD growth of graphene films, one of the most important challenges is to prevent poor-quality carbon-rich film deposition prior to the actual graphene CVD growth. It was found that the deposition time, temperature regimes, and atmosphere play a crucial role in controlling the number of layers deposited on the foils [30]. Another important parameter for the high-quality graphene growth by CVD is the quality of the substrates [29]. Since grain boundaries act as nucleation sites for graphene, the growth of the morphology of the metal film has a major effect on the graphene growth process. To achieve a defined surface state, thermal treatment of the metal substrates in a hydrogen atmosphere prior to the growth stage has been proposed [31]. The chemical nature of hydrogen is believed to be sufficient to reduce and thereby remove carbon residues as well as native oxides.

### 2.2. Graphene via Chemical Method

The chemical method is relatively simple with a higher yield, which makes it popular for research-scale graphene production [25,28,32]. Hummer’s method is the most used pathway, and the raw graphene obtained by this method is GO. This method commonly involves the oxidation and exfoliation of multilayered graphite to single (or multi-) layered GO, using strong oxidizing agents such as potassium permanganate (KmnO4) [33]. Due to the presence of oxygen-containing functional groups such as -OH, -COOH attached to graphene layers, GO exhibits good hydrophilicity and hence excellent dispersion in numerous solvents, unlike graphene and graphite powder. However, the method suffers from poor product quality due to the formation of incomplete oxidized graphene/graphite particles as final products. For quality improvement, modifications of Hummer’s reaction were proposed by different scientists. Hummers and Offemann et al. presented an optimized, rapid, safe method of graphite oxidation, whereby GO is obtained via the treatment of graphite by strong oxidizing agents [34,35]. In this method, the graphite was pre-oxidized by a mild oxidizing agent such as sodium nitrate (NaNO_3_) or a strong oxidizing agent such as a mixture of phosphorus pentoxide (P_2_O_5_) and potassium persulphate (K_2_S_2_O_8_) in sulphuric acid as the first step. Besides the oxidation methods that affect the physicochemical properties of the final products, the literature reports [36,37] that the resulting GO structure depends on the type of graphite and its structural parameters—mainly the size of the crystallites.

GO can then be significantly restored or reduced to graphene sheets by chemical reducing agents or thermal annealing, which is generally known as reduced graphene oxide (rGO). During the reduction process, oxygen-containing functional groups are removed from GO, which restores oxidation defects and long-range conjugation over the graphene network [25]. Despite its popularity, most of the time, using chemical methods results in products that have electrical properties and surface area that are below expectations. The highest degree of reduction and the smallest number of structural defects were obtained for rGO prepared using flake graphite, whose crystallite diameter was the largest among the examined graphite materials [38]. The rGO obtained from scalar graphite showed the smallest degree of reduction and share of carbon with C sp2 hybridization and the highest degree of structural defects [39]. It is also worth mentioning that the largest surface area values (over 900 m^2^ g^−1^) were measured for rGO obtained from scale and flake graphite and oxidized using a modified version of Tour’s method [40]. X-ray diffraction (XRD), X-ray photoelectron spectroscopy (XPS) and Raman spectroscopy (RS) are among various other tools to analyze the rGO structural parameters and oxidation degree.

Even though the oxygen content may be the same, GO and rGO show different electron conductivity, oxidation and absorption abilities. These differences in property arise from the variations in the underlying graphitic structure, degree of oxidation and the type of defect present.

### 2.3. Exfoliation of Graphite

The exfoliation of graphite to achieve graphene is one of the most promising ways to achieve large-scale production at a very low cost [41,42,43,44]. Exfoliation is a top-down approach that requires mechanical energy to exfoliate graphite. Although this production method is simple enough, the quality and purity of the exfoliated graphene may not be sufficient for specific applications. Mechanical exfoliation of graphite results in either a stack of sheets or a small number of detached sheets. This depends on the condition of mechanical exfoliation. The end products are usually FLGs and GNPs or a mix of the two.

FLGs and GNPs combine large-scale production and low costs with remarkable physical properties [45], including electrochemical performance for energy storage applications [46,47]. These nanoflake powders are normally obtained following a liquid-phase exfoliation and provide certain health and safety benefits over other types of GBM or nanocarbons [48]. Compared to GO, less-oxidized FLG and GNP are more conductive and, in general, are easier to mass-produce [49]. Other widespread FLG and GNP manufacturing methods include the exposure of acid-intercalated graphite to microwave radiation, [50] shear-exfoliation, [51] and the more recent wet-jet milling. These manufacturing techniques produce a large variety of powders in terms of thickness, the lateral size of the flakes, aspect ratio, and defect concentrations. Commercially available graphene is typically a mixture of FLG and GNP.

The potential of graphene for Li-ion batteries has been significant as demonstrated in various works. In general, the role of graphene is to offer directional pathways for electrons and Li ions to enhance the electronic and ionic conductivity of electrode materials. In electrolytes, GO has been used for the purpose of enhancing Li ionic conductivity, mechanical strength, thermal stability, and fracture toughness. Further, graphene or rGOs improve interfacial properties when they are used as interlayers at the electrodes|SSE junctions.

## 3. Graphene-Enhanced Solid Electrolytes

### 3.1. GO as Randomly Oriented Fillers in SPEs

Among various SSEs, SPEs generally offer lower ionic conductivity and mechanical strength than their ceramic counterparts. The properties of SPEs can be tailored by the incorporation of different inorganic particles in their formulation [13,14,52,53,54]. By Lewis acid-base interactions of the surface-functionalized inorganic particles with the Li-containing polymer matrix, better dissociation of Li salt is achieved, which helps to improve the ionic conductivity [55,56]. Moreover, some nanoparticles are exclusively used to control the crystal growth rate and final crystallinity of the SPEs [52]. Further, the uniform distribution of the rigid ceramic particles in polymer matrices can enhance the mechanical strength.

GO plays a similar role when used as a nanofiller in SPEs. Firstly, various functional groups (single bond OH, single bond COOH, single bond C double bond O, single bond COC, etc.) on its edges and basal planes can favor Li salt dissociation [57] and hence increase the conductivity. Secondly, by making a mechanical interlocking/adhesion with polymer, GO reinforces SPEs to achieve better mechanical properties [58]. Lastly, GO improves the thermal properties of the electrolyte thanks to its high thermal stability. Several works in the literature have reported the use of GO instead of rGO in SSEs [57,59,60,61,62,63,64,65,66,67,68]. GO is preferred because of its electronically insulating nature, which lowers the possibility of electronic leakage or short-circuits in the SSE.

**Figure 2 nanomaterials-12-02310-f002:**
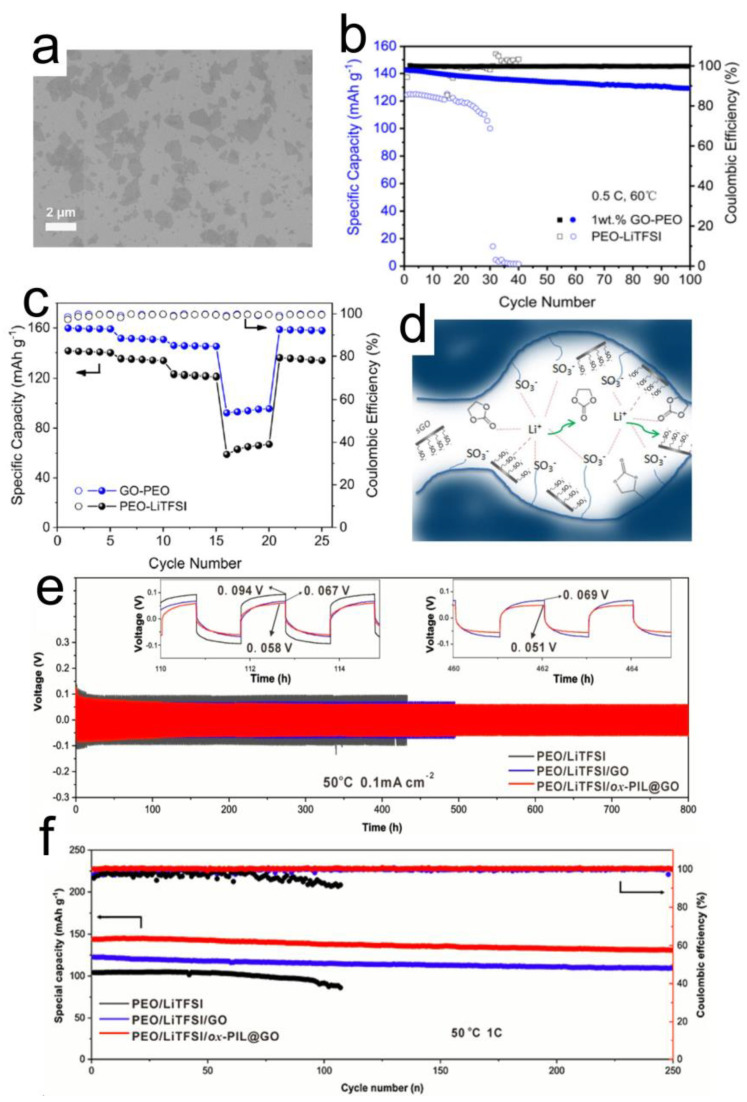
(**a**) GO nanosheets are dispersed inside the polymer matrix; (**b**) Cycle life tests (**c**) Rate performance of cells with and without GO filler in Poly(ethylene oxide (PEO). Reprinted with permission [59]. (**d**) Schematic illustration of Nafion nanocomposite incorporated with sulphonated GO. Reprinted with permission [69]. Electrochemical performance comparison of SPEs based on only PEO, PEO-GO and PEO- ox-PIL@GO (**e**) Li symmetric cell cycling at between 0.1 mA cm^−2^ and 50 °C. (**f**) Full cell LiFePO_4_|CPE|Li at 1 C- rate and 50 °C. Reprinted with permission [70].

In a recent work [59], it was shown that adding 1 wt% GO to the PEO polymer matrix brings a seven times increase in room temperature ionic conductivity and lowers the activation energy from 1 to 0.64 eV of the electrolyte. GO improved the ionic conductivity by lowering the crystallinity of the PEO polymer via suppressing the formation of crystalline nuclei and thus increasing the segmental motions of the individual chains. As shown in Figure 2a, GO sheets have been dispersed in the polymer matrix. A Li symmetric cell based on the SSE provided stable cycling for more than 600 h with ~0.027 V of overpotential. The Li|LiFePO_4_ (LFP) full cell delivered an initial discharge capacity of 142 mAh g^–1^ at 0.5 C and 91% capacity retention after 100 cycles (Figure 2b). Notably, the cyclic performance of the PEO-GO sample is much better than that of neat PEO. The enhancement in conductivity translated into better C-rate performance as shown in Figure 2c. Polyacrylonitrile (PAN) SPEs have also benefited from GO as a nanofiller [62,71]. Similar to PEO polymers, here the weight ratio was kept very low (~1%). The impact was observed in the form of increased room temperature ionic conductivity and improved mechanical strength. It was argued that the functional groups on GO decreased the polarity of C-N, helped in the dissociation of Li salt (LiClO_4_) and made the polymer softer [62]. A comparison of LFP cells’ performances demonstrated a 166 mAh g^−1^ at 0.2 C-rate for GO-based electrolyte that was higher than the cells without GO (136 mAh g^−1^). The use of GO has also been reported in another work where poly(ethylene glycol)-grafted GO (PEG-GO) is incorporated in PEG methyl ether methacrylate polymer electrolytes [72]. Here too, GO helped in increasing the Lewis acid-base interaction with Li salt, which resulted in its higher dissociation to finally achieve enhanced ionic conductivity at room temperature. A rather different approach was reported where instead of adding GO as a filler to Nafion SPE, an ion-exchange reaction in LiOH solution was carried to lithiate GO [69]. The Nafion nanocomposite incorporating sulphonated graphene oxide (sGO-Li^+^) was finally swelled with organic solvents that acted as the SPE. The authors argued that molecular complexes of Li^+^ ions are crosslinked with sulfonic groups (Nafion and/or sGO) and carboxylic oxygens of the solvents that allow Li^+^ ions transport via a segmental motion of the polymer chains and the solvent molecules, as shown schematically in Figure 2d. Overall, the presence of sGO favors the formation of an appropriate network that promotes ion transport, resulting in higher ionic conductivity, Li reversibility, and improved electrochemical response.

Bao et al. [70] claimed that an issue pertaining to SPEs with GO as nanofiller is the random agglomeration and leakage of GO, giving rise to Li dendrite growth. They proposed functionalized GO based on oxyethyl-containing poly(ionic liquids (ox-PIL@GO) as nanofiller. Due to coordinated, electrostatic and ion-dipole interactions among PEO matrix, LiTFSI and oxPIL, better distribution of GO is achieved, which helps in reducing the crystallinity of the electrolyte uniformly. As a result, the ox-PIL@GO incorporated SPEs outperform SPEs without ox-PIL or non-functionalized GO. As shown in Figure 2e, the Li|Li symmetric cell runs much longer without a dendritic short-circuit. Furthermore, the LFP|CPE|Li cell delivers a much higher capacity and cycle life (Figure 2f).

### 3.2. Dimensional and Aligned GO as Fillers in SPEs

In most of the works reported so far, the GO is randomly oriented in the polymer matrices, which has a beneficial impact on the performance of the SSBs as shown in the previous section. However, aligning GO nanostructures inside the polymer matrix could potentially improve the performance further. Few works have focused on this topic [73,74,75,76,77]. For example, Cheng et al. [74] showed that aligning 2D GO nanosheets inside the PEO polymer results in higher room temperature ionic conductivity. In such a structure, the aligned GOs play the dual role of confining PEO crystal orientation as well as crystallization. They did not use any complex synthesis process but rather included a simple step of slowly evaporating the solvent to achieve the orientation. The 2D wide-angle XRD pattern was observed in-plane (Figure 3A) for GO-PEO samples but not through the plane. This confirms that GO nanosheets are oriented parallel to the film surface. In such a structure, the ion transport is guided by GO nanoplatelets and the oriented PEO lamellae, leading to high anisotropic ionic conductivity. Another study also focused on controlling the filler direction in GO-polymer composite electrolytes [75]. While it is known that GO nucleates polymer crystallization, polymer crystallization also has an impact on the microstructure of GO. It was observed that the interfacial crystallization of PEG on GO improves both GO alignment and the orientation of PEG crystal within the polymer matrix, which should be of significant interest in designing high-performance SPEs.

In a recent paper, Zhou et al. [76] took it one step further by developing a 3-dimensional GO aerogel foam filled with PEO polymer. The GO aerogel film was freeze-dried in a mold beforehand. Figure 3B and C show SEM images of the GO framework before and after PEO filling, respectively. A significantly improved cycling behavior was observed for the 3D GO structures in comparison to neat PEO samples, especially at higher (50 °C) temperatures (Figure 3D). The authors attributed such improved performance to the improved Li^+^ ion mobility via ordered channels provided by the 3D graphene. Furthermore, the skeleton structures were responsible for the mechanical integrity of the electrolyte that helped in suppressing Li dendrites.

### 3.3. GO Role in Flexible Thin Film Battery

There is a lot of interest in flexible electronics, such as roll-up mobile phones, flexible watches, smart garments, and so on [78,79,80]. Flexible and sleek batteries will be used to power such devices. Such batteries will not only be required to perform well but also to comply with structural deformations, such as bending, stretching, and twisting, which makes SPE a promising candidate for such devices. Here, it is vital that the SPEs are mechanically robust to withstand the mechanical deformations without compromising the electrochemical performance. Polymer composites with GOs have been shown to fulfill such requirements thanks to their high mechanical strength [59,62,81,82]. Kammoun et al. developed a polymer nanocomposite electrolyte composed of 1 wt% GO in a PEO matrix [81]. While reasonable electrochemical performance in terms of maximum operating voltage (4.9 V) and energy density (4.8 mW h cm^−3^) at room temperature was observed, what struck was the mechanical endurance of the electrolyte. As shown in Figure 4a, 93% of the operating voltage is maintained after 6000 bending cycles. Tensile tests confirmed a strong electrode|electrolyte adhesion. The comparison showed that batteries with 1 wt% GO outperform those without GO as shown in Figure 4b. Further, it was demonstrated that when in bent form, the GO-PEO electrolytes perform even better than when in a flat position. The possible reason could be that the excessive stress created on electrode|electrolyte contact in a bent position helped achieve a better contact. This agrees with some other reports where pressure has been an important parameter to enhance solid–solid interface contact between Li and SSEs [83,84,85]. Wen et al. also observed improvements in tensile properties and enhanced flexibility by adding 1 wt% GO to a polymer electrolyte [59]. With such mechanical traits, the CPEs would alleviate malfunctions due to the stresses induced by mechanical deformations.

In a recent work, a flexible SSB based on GO-modified poly (vinylidene fluoride-tri-fluoroethylene-chlorofluoroethylene) [PTC] polymer was demonstrated [82]. The electrolyte membrane was coated directly on top of the electrodes using the electrospinning technique, then swollen with liquid electrolyte and subsequently vacuum dried to remove all the solvents. The GO was responsible for the improvement of the thermal and mechanical properties of the film when compared with samples without GO or the conventional Celgard separator. In terms of ionic conductivity, the GO-enhanced electrolyte was able to operate in a wide range of temperatures (−15 °C to 160 °C). A full cell-based on LCO cathode, GO-PTC SPE, and graphite anode was tested under mechanically deformed conditions. As shown in Figure 4c, the cell OCV was maintained rather well for more than 1000 cycles under both bend and flat conditions, suggesting good flexibility and mechanical robustness of the electrolyte. For the electrochemical performance of the flexible LCO/graphite full cell using the GO-PTC composite electrolyte, while the room temperature capacity was slightly lower (~120 mAh g^−1^), but it was stable for more than 100 cycles (Figure 4d). The capacity improved at higher temperatures (45 °C, 60 °C, and 80 °C), although capacity decay was more pronounced at 80 °C possible due to accelerated side reactions [86,87,88]. To showcase its safety features, the flexible battery powered an LED bulb under various conditions (flat, rolled, and cut) as shown in Figure 4e. Furthermore, the cell was able to power an Apple watch, demonstrating its feasibility for wearable applications.

### 3.4. GO in Ceramic SSEs

While there has been a lot of focus on using GO to improve the thermal, mechanical, and electrochemical properties of solid polymer electrolytes, not much has been reported on their use in ceramic SSEs. This may be partly due to the reason that mechanical or thermal issues are not that concerning in ceramic systems compared to SPEs. Nevertheless, when used in a single piece (as a disk separator/pellet), ceramic SSEs do face problems such as low fracture toughness, especially when operated under stack pressures. Any tiny microscale fracture in the bulk of electrolyte provides paths for Li dendrites to sneak through that poses a risk of cell short circuit [89,90]. It is therefore desirable to increase the fracture toughness of such electrolytes. The use of graphene (rGO) has been shown as a nanoscale reinforcer to increase the toughness of various SSEs [91,92,93,94]. Athanasiou et al. incorporated low volume fractions (1–5 vol%) of rGO in a Li-ion conducting LATP SSE [94]. It was demonstrated that the mean toughness value for LATP increased from 1.1 MPa.m^0.5^ for 1 vol% to 2.4 MPa.m^0.5^ for 5 vol% (Figure 5a). It was argued that the increased fracture toughness was due to the crack deflection, bridging, and pull-out of the rGO platelets, as schematically represented in Figure 5b. Qualitative evidence was provided by SEM images (Figure 5c,d), where rGO can be seen bridging and pulling out at the micro cracks. It is worth mentioning that there is a risk associated with the use of rGO in the form of high electronic conductivity. The authors were aware of this, which is why they tried different amounts of rGO in the composite. It was shown (Figure 5e) that the electronic conductivity of the electrolyte is higher than its ionic conductivity when 5 vol% rGO is used, possibly due to the creation of a percolating conductive network through the materials. They concluded that an amount of 1 vol% is optimum with negligible electronic conductivity but enhanced toughness. The electrochemical response for the SSE was recorded in Li symmetrical cells. As shown in Figure 5f, the overpotential stays constant over 250 h of cycling, suggesting no increase in the interfacial resistance. The work is important since it can create more research interest to explore the role of graphene in enhancing the crack toughness of various other types of ceramic SSEs such as LLZO, LGPS, LPS, and so on.

Another work explored a different role of GO in a sulfide LPS SSE [95]. Due to the electrochemical instability of sulfide electrolytes with Li metal, it is advantageous to coat the materials with protective layers. It was shown that coating a small amount of GO (1 wt%) on Li_7_P_3_S_11_ SSE particles results in a stable and efficient all-solid-state battery performance. The synthesis of GO@LPS materials was performed using a simple facial technique where GO and LPS were mixed in acetonitrile followed by an annealing step as shown in Figure 5g. The surface-modified powder of LPS was pressed into SSE disks and then integrated with the Li anode and LCO cathode. Upon reaction with Li^+^ ions, GO is reduced to rGO; however, since a very small amount is added (~1 wt%), no percolation network is established. This was confirmed when the electronic conductivity of the electrolyte was 5 orders of magnitude lower than the ionic conductivity. Stable Li stripping/platting voltage profiles were observed for more than 600 h, unlike the uncoated LPS that short-circuited even before reaching 50 h of cycling. The Li|GO@LPS|LCO cells performed much better than cells based on only LPS electrolyte as shown in Figure 5h. This work highlights the advantage of GO in preventing LPS SSE from directly contacting Li metal and at the same time regulating Li^+^ ion flux at the interface.

## 4. Electrodes in an SSB Based on GBM

### 4.1. Graphene in Cathodes

GBM-enhanced cathode materials have performed well when integrated with different SSEs. Among various cathodes, sulfur is a popular choice due to its high theoretical capacity of 1672 mAh g^−1^. It is well known that sulfur as an electrode in liquid electrolytes is problematic due to the dissolution and “shuttle effect” of polysulfides, which greatly decrease cycle life and specific capacity. Replacing the liquid electrolyte with the SSEs can somehow address the polysulfide dissolution, and improve the safety and cycle stability. However, integrating sulfur cathodes with SSEs can increase the cell resistance due to poor electrode|electrolyte contact, although, some solutions have been found for this problem [96]. In addition, the insulating characteristics of sulfur and its discharge products are another challenge to be overcome [97]. The ionically and electronically insulating active material requires composite formation with electron-conductive additives to secure sufficient ion and electron supply. Therefore, intensive efforts have been devoted to addressing the mentioned issues. Graphene is expected to be a promising component in preparing composite electrodes with high electronic conductivity. Enhanced electron flow is achieved by establishing a percolation network, which provides electronic conduction pathways [98,99]. Generally, parameters such as aspect ratio, conductivity, orientation, dispersion, and concentration of conductive filler influence the percolation threshold. GMBs, with a higher aspect ratio, have more probability of contacting each other; therefore, presenting a lower percolation threshold.

To increase the electrical conductivity of the sulfur cathode, Xiayin et al. [100] proposed a composite of rGO-sulfur by the deposition of crystal and amorphous sulfur on rGO. As discussed earlier, rGO is the more conductive form of GO with fewer functional groups. The reduced oxygen content compared to GO makes it more conductive but less suitable for coating. To achieve high ionic conductivity, rGO@S nanocomposite was uniformly distributed into mixed conducting LGPS-acetylene black composite. Since the LGPS electrolyte is not stable with the lithium anode, a lithium-compatible 75%Li_2_S-24%P_2_S_5_-1%P_2_O_5_ electrolyte layer was inserted between the LGPS electrolyte layer and the lithium anode to avoid any undesired reactions between lithium metal and LGPS (Figure 6a). At 60 °C, the all-solid-state Li–S cell demonstrated high discharge capacities (~1100 mAh g^−1^ at 1.0 C) (Figure 6b) and long-range cycling stabilities with a retention of 830 mAh g^−1^ at 1 C for 750 cycles. The reason behind the poor performance of the amorphous rGO@S composite at 25 °C and 100 °C was studied by EIS. The high ionic resistance of Li–S cells at 25 °C was due to the poor ionic conductivity of SSEs at this temperature. The poor performance of the electrolyte bilayer at 100 °C was attributed to the electrolyte decomposition. This cathode showed excellent rate capability due to the close contact between sulfur and the graphene membrane. In addition, the uniformly dispersed rGO@S into the electrolyte generates an even volume change in the cathode and extends the cycling stability.

In another work [101], a sulfur/graphene nanosheet (S/GNS) composite (with a weight ratio of 3:1) was prepared through ball-milling. The Li-S/GNS cell is separated by the PVDFHFP/PMMA/SiO_2_ gel polymer electrolyte (GPE) film. The cell exhibited a high initial specific discharge capacity (809 mAh g^−1^) and maintained a reversible discharge capacity of 413 mAh g^−1^ after 50 cycles under a 0.2 C rate. The cycling performance of the cell indicates that the combination of the S/GNS composite cathode and GPE plays a significant role of retarding diffusion of the polysulfides out of the cathode and suppressing their transport towards the anode side. The rate capability performance was also investigated, and the results indicated that the cell delivers a reversible discharge capacity of 638 mAh g^−1^ at 0.1 C. The high-rate operation was ascribed to the enhanced conductivity of the GNSs, which act as nano-current collectors, and the good ionic conductivity of the GPE.

**Figure 6 nanomaterials-12-02310-f006:**
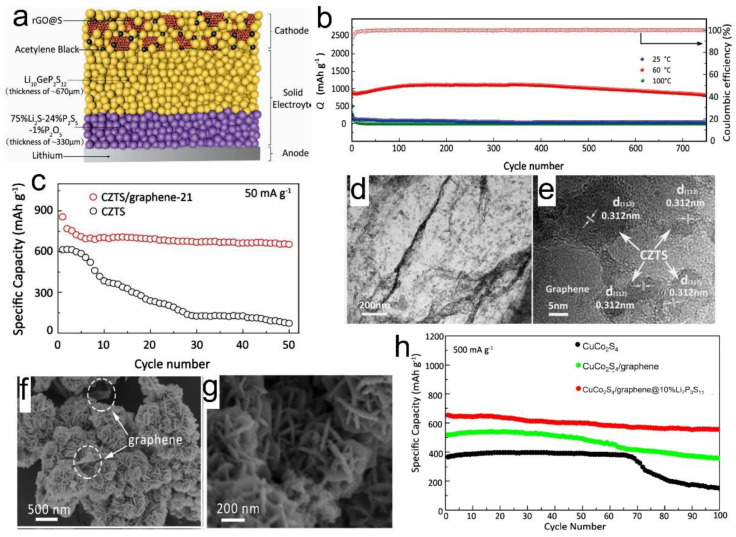
(**a**) Schematic diagram of an all-solid-state lithium–sulfur battery; (**b**) Cycling performances of amorphous rGO@S-40 composites under the high rate of 1 C and corresponding Coulombic efficiencies at 60 °C. Reprinted with permission [100]. (**c**) Cyclic performances for pure CZTS and CZTS/graphene under the current density of 50 mA g^−1^ at room temperature; (**d**,**e**) TEM and HRTEM images of the CZTS/graphene. Reprinted with permission [102]. FESEM images of (**f**) CuCo_2_S_4_/graphene, and (**g**) CuCo_2_S_4_/graphene@10%Li_7_P_3_S_11_ samples; (**h**) Cycling performances of pure CuCo_2_S_4_, CuCo_2_S_4_/graphene, and CuCo_2_S_4_/graphene@10% Li_7_P_3_S_11_ electrodes in all-solid-state lithium batteries at the current density of 500 mA. Reprinted with permission [103].

Kızılaslan et al. investigated the effects of two different synthesis methods of rGO-S composites cathodes, including the melt-diffusion method and solution-based techniques [104]. They argued that the composite cathode prepared through the solution-based technique results in cells that display the highest capacity of 1010 mAh g^−1^ and favorable cycling performance of 740 mAh g^−1^ after 200 cycles. The interaction between the rGO and sulfur not only enhanced electronic conduction in the cathode but also suppressed the volume change during the conversion reaction.

Wan et al. [102] designed a graphene-based composite cathode (Cu_2_ZnSnS_4_/GO) via a hydrothermal approach. GO could deposit and adhere to the surface of the materials due to its hydrophilic nature as it contains more oxygen contents as compared to other forms of graphene. The results revealed a discharge capacity of 645 mAh g^−1^ after 50 cycles (Figure 6c) for Cu_2_ZnSnS_4_/graphene (CZTS/graphene) with a weight ratio of 2:1 using a bilayer solid electrolyte (LGPS and 70%Li_2_S-29%P_2_S_5_-1%P_2_O_5_). Furthermore, a good rate of performance was observed where discharge capacities ~527, ~439, and ~278 mAh g^−1^ were delivered at 100, 250, and 1000 mA g^−1^, respectively. TEM and HRTEM images, as shown in Figure 6d,e, confirm the morphologies and structures of CZTS/graphene samples. It is evident that GO plays a beneficial role in forming intimate contacts between CZTS nanoparticles and sulfide electrolytes, thus constructing favorable ionic and electronic conduction pathways.

In another work, CuCo_2_S_4_, CuCo_2_S_4_/graphene, and CuCo_2_S_4_/graphene@Li_7_P_3_S_11_ nanocomposites were studied [103]. To synthesize these cathodes, GO and CuCo_2_S_4_ aqueous solutions were mixed and dried via hydrothermal processes, resulting in composites as shown in Figure 6f. Then, Li_7_P_3_S_11_ was coated on the surface of CuCo_2_S_4_/graphene nanosheets (Figure 6g). The cell exhibited a reversible capacity of 556 mAh g^−1^ at a high current density of 500 mA g^−1^ after 100 cycles Figure 6h. The excellent electrochemical performance of CuCo_2_S_4_/graphene@Li_7_P_3_S_11_ was attributed to the high electronic conductivity of Cu/rGO and the high ionic conductance of LGPS-acetylene electrolyte, ensuring fast electron and ion conductions. Moreover, graphene helped to alleviate electrode volume changes during repeated charging/discharging.

Few works have shown GBM enhancements in transition metal oxide-based cathodes [105,106]. For example, Dai et al. employed graphene fluoride as a conductive agent for the LCO cathode composite to alleviate the undesirable decomposition reactions at the electrolyte interface [105]. The LiF-rich layer helped in providing chemical stability to the Li-argyrodite SSE, while the graphene ensured better contact and improved electronic conduction in the electrode.

### 4.2. Graphene in Anodes

As mentioned previously, Li is an ideal anode material for SSB due to its favorable electrochemical properties. However, problems caused by Li dendrites, such as low coulombic efficiency and short circuits, have limited their practical application for Li-metal batteries. In this regard, Li stability with the electrolyte is very important and many works have focused on the interface modification of the Li anode. Equally important is the Li metal and current collector interfacial stability. The current collector works as an electrical conductor between the electrode and the external circuits as well as a substrate for electrode coating. Commercially, copper (Cu) is a preferred current collector for Li anode due to its electrochemical stability at low potentials (0.01–0.25 V vs. Li/Li^+^ voltage). To improve Li anode performance, the incorporation of novel nanostructured materials such as graphene with Cu is being actively considered [107,108,109]. The CVD-derived graphene on Cu mesh (Figure 7a,b) improved the reversible capacity of Li-based SSBs by lowering the impedance [110]. As a result, the modified current collector @Li||NCM showed capacity retention of 84% and could still stably operate for 150 cycles at a high current density of 5 mA cm^−2^ (Figure 7c). This was a significant improvement over the bare Cu foil, which exhibited a rapid decay within 30 cycles, which was due to the irreversible loss of active Li metal and continuous Li dendritic growth. Line-scan EDS and XPS profiles confirmed the uniform nucleation and growth of the Li on the modified current collector.

In another work, the role of rGO is demonstrated in a composite anode with a ceramic Argyrodite Li_6_PS_5_Cl SSE [111]. The 2D rGO not only acted as a template to achieve Li_6_PS_5_Cl particles with a high aspect ratio but also helped in enhancing the electronic conductivity of the electrode. The composite had a good distribution of rGO and the SSE, as shown in the cross-sectional SEM and EDS images in Figure 7d. Percolation networks were established in such a structure that improved both the ionic and electronic conductivity of the electrode. Improvements were evident in the battery performance where composite anodes based on rGO (G), Li_6_PS_5_Cl (L) and natural graphite (NG) demonstrated better rate capability and capacity retention than electrodes without rGO (Figure 7e). NG:GL (1:80) performed the best out of all the samples.

Flexible SSBs will require the development of not only flexible SSEs but also thin and flexible electrodes [113,114,115]. The role of graphene is important in realizing flexible anodes. As demonstrated in the work [116], a mechanically flexible anode was obtained through depositing CVD of monolayer graphene directly onto the copper foil. The SSB with an area of 1 mm^2^ and a thickness of ∼600 nm was fabricated by lithography. The cell exhibited a charge capacity of ∼7 µAh cm^−2^ after 10 cycles. In another work, a bendable all-solid-state battery (∼50 µm thick) was created by using monolayer graphene grown on a Cu foil as the electrode (∼25 µm thick) and integrated with poly-(ethylene glycol) borate ester thin SPE [112]. The practical application of the cell was demonstrated (Figure 7f), where it was able to power an LED in an extremely bent form.

## 5. Graphene Role as an Interface Mediator at the Electrode/SSE Interface

With SSEs, the focus has been on enhancing the room temperature ionic conductivities of the electrolyte to reach or even surpass those achieved for conventional organic liquid electrolytes. There has been success on this front, especially with ceramic electrolytes where room temperature ionic conductivities >10^−2^ S cm^−1^ have been reported [9,117,118,119]. With good ionic conductivities achieved, the next goal is to integrate Li anodes with SSEs to build batteries delivering the highest possible energy density on a cell level. However, due to the nature of the solid–solid interface, a mechanical and electrochemically stable interface is hard to achieve in SSBs [96]. The electrode|electrolyte interfacial kinetics of Li^+^ ions in a solid-state system depends on the following two important factors: firstly, the (electro-)chemical stability at the interface; and secondly, the contact intimacy and mechanical robustness. Once Li contacts the SSE, interphase layers start to form due to Li reaction and electrolyte decomposition [117,120]. Interphase layers with a finite thickness (tens of nanometers) are generally considered stable since they provide sufficient passivation to the electrode and prevent further electrolyte decomposition. If passivation is not achieved, the electrolyte will continue to decompose, which will result in non-uniform and thicker interphase layers offering enormous resistance to the Li^+^ ion transport [121,122]. Further, insufficient Li|SSE contact can also result in higher interfacial resistance and low utilization of electrode capacity. In the case of Li metal as an anode, the issue is even more severe due to the excessive Li expansion and contraction (as large as tens of micrometers) that makes it harder to cycle SSBs at high current densities. Mechanical stresses are more pronounced due to the continuous change of lattice of electrodes during Li (de-) insertion, which may delaminate the electrodes from the electrolyte. Among various strategies, introducing interlayers at the electrode|SSE interface is a viable approach to solving the interfacial problems. Numerous works have demonstrated that using polymer, liquid, and/or metal-oxide interlayers helps achieve better electrode|SSE contact resulting in an improved electrochemical performance [123,124,125,126]. In line with such approaches, graphene has also the potential to be used as an interlayer material at the electrode|SSE junction. This is mainly possible because of the favorable traits of graphene, such as mechanical robustness, structural flexibility, and reactivity with Li^+^ ions to form alloys that would ensure stable interfacial contact.

Several reports demonstrate the use of graphene or GBMs at the electrode|SSE interfaces via simple facial approaches [95,127,128,129,130]. Generally, the idea is to coat a thin layer on the planer surface of either the Li anode or the cathode film. Zhang et al. reported the in situ formation of a LiF/graphene inorganic composite layer at the LFP|LLZO interface [128]. In a typical process, fluorinated graphene (GF) was coated via facial methods on the LLZO surface and subsequently dried. During prelithiation (GF + Li^+^ → LiF + Gr), GF was converted to graphene and LiF that acted as an interlayer between LFP and LLZO, as shown in Figure 8a. Without the interlayer, the cell impedance was much high, as shown in Figure 8b. Interestingly, after lithiation, the resistance decreased even further, highlighting the importance of graphene reactivity with Li^+^. Since LLZO is very brittle in nature, achieving an intimate electrode|SSE contact is always an issue. The authors claim that a flexible graphene/LiF interlayer ensures an intimate contact that imparts stability to the Li^+^ ion transport; consequently, an improved battery performance (140 mAh/g after 60 cycles) is achieved than cells without interlayers (Figure 8c). The formation of an in situ graphene layer has also been demonstrated in another work [131]. Here, by making GO react with the Li surface (2Li + GO + 2H^+^ → 2Li^+^ + rGO + H_2_O) and subsequent coating of poly(propylene carbonate) results in the formation of an in situ rGO that acts as an interlayer. The GO-modified electrolyte showed good properties such as a wide electrochemical window (up to 4.8 V), ionic conductivity (~2 × 10^–4^ S cm^–1^), and a high ionic transference (0.9). When the interface-modified SPE was paired with Li metal and NCM622 electrodes, an initial specific capacity of about 160 mAh g^−1^ at 0.5 C was achieved. While there was some capacity decay, nevertheless, ~100 mAh g^−1^ was still maintained after 200 cycles. GO has also been used as a thin interlayer between Li and sulfide SSEs [95]. Here, the main advantage of GO was to isolate Li_7_P_3_S_11_ (LPS) SSE from directly contacting Li metal, which otherwise would cause undesired reactions. In another work, coating GO thin layers on LiNi_0.5_Mn_0.3_Co_0.2_O_2_ (NMC532) and then integrating with a poly(propylene carbonate) based SPE helped retard the decomposition of PPC electrolyte that may be induced by the oxidized species (Ni^3+^ and Co^4+^) during the charging cycle of the cathode [106]. The improvements in the electrochemical performance are presented in Figure 8d, where GO@NMC shows much lower capacity decay.

Graphene has also been demonstrated to help achieve better contact in a 3D architectural Li metal anode [132]. Existing strategies involve planar Li foil, but considering the enormous volume change associated with Li, there is every possibility that it could delaminate from the SSE under high current testing. A different approach was adopted where metallic Li in layered reduced graphene oxide host (Li-rGO) was used as the anode, and viscous semiliquid PEG was impregnated into the 3D Li-rGO via thermal infiltration to construct the flowable interphase as represented in schematic Figure 8e. Battery assembly was performed by integrating the 3D Li-rGO electrode with PEG polymer and LLZO ceramic SSE. Li strip/plat tests for the Li-rGO symmetric cells demonstrated stable voltage profiles with low voltage polarizations for extended hours of testing, suggesting efficient suppression of Li dendrites (Figure 8f). The full cell comprising Li-rGO anode, PEG electrolyte, and LFP cathode demonstrated much better rate capabilities and cycle life than the cells without 3D Li-rGO architecture (Li bare foils) (Figure 8g). The authors argued that the surface functional groups on rGO make its surface “lithiophilic” towards molten Li, and the nanoscale gaps between the rGO layers can provide a strong capillary force to drive the molten Li intake into the rGO host. A few reports have also shown the use of graphite to enhance the electrode|SSE interfaces [129,130]. In each case, the improved interfacial performance is due to the graphite individual layers (graphene) that help in forming intimate contact with metallic Li and creating fast Li^+^ conductive layers of LiC_6_, thus facilitating the uniform deposition of Li and inhibiting Li dendrite formation during long-term cycling.

## 6. Summary and Prospects

In this review, we have presented a summary of the recent progress in SSBs, with a focus on the role of GBMs in enhancing their electrochemical performance.

Graphene exhibits novel electrochemical properties such as low charge transfer, wide potential window, excellent electrochemical activity, and fast electron transfer rate that could be useful for many applications, including LIBs. The synthesis involves chemical and/or electrochemical exfoliations of graphite to form a monolayer, FLG, or GNP graphene platelets optionally containing various functional groups. GBMs are particularly suitable since they are beneficial for each component (electrolyte, electrode, and interface) of an SSB as summarized in Table 1.

For SSEs, GO has been the most studied type so far due to its electronically insulating nature, hence excluding any possibility of electronic leakage or short-circuits, at the same time enhancing conductivities and improving mechanical and thermal properties. Ionic conductivity is improved by facilitating Li salt dissociation. Further, GOs can mechanically interlock with polymeric chains thanks to their 2D geometry, which reinforces the SPEs to achieve better mechanical properties. Lastly, SPEs based on GO can withstand higher temperatures thanks to their high thermal stability. In most of the works, the SSEs are composites of randomly oriented GOs and polymers. However, a few works have also explored the potential of using 3D GO as filler. In such structures, improved performances are observed due to enhanced Li^+^ ion transport via ordered channels provided by the 3D GO. The high mechanical strength imparted by GO has also encouraged the application of CPEs for flexible batteries. Furthermore, rGO has benefited ceramic SSEs by enhancing their fracture toughness, which could potentially be useful for inhibiting penetration of Li dendrites. It is worth mentioning that compared to less oxidized forms of GBMs, GO has various safety and regulatory hurdles that limit large-scale production. Therefore, the main applications for SSBs are mostly demonstrated at a lab-scale level.

The role of GBMs in both cathodes and anodes is also of significant importance. Here, some of the advantages of rGO are to provide a large surface area for Li intercalation, offer high electronic conductivity to lower the electrode resistance, and suppress the agglomeration and deformation of the electrode nanostructures during cycling. In the case of the sulfur cathode, graphene can help to retard the diffusion of the polysulfides out of the cathode and suppress their shuttle towards the electrolyte and the counter electrode. Hence, not only inhibiting active material loss but also minimizing anode corrosion. They can also enhance particle-to-particle contact in transition metal oxide cathodes. On the other hand, graphene’s benefits for the anode are also substantial, such as its use as thin-film electrodes to store charge for miniature flexible thin-film batteries, with a Cu current collector to ensure less resistive Li|Cu contact, and as a template to achieve 2D morphologies for ceramic SSEs in a composite.

Alongside their role in SSEs and electrodes, graphene has also the potential to be used as an interlayer at the solid–solid interface at the electrode|SSE junctions. This is mainly possible because of favorable traits of graphene such as mechanical robustness, structural flexibility, and reactivity with Li^+^ ions to form alloys that would ensure an electrochemically stable and mechanically intimate electrode|SSE contact.

The future is going to see an enormous demand for batteries with high storage capacity and faster charge rates. FLG and GNP with low oxygen content are becoming more commercially available and affordable and might offer a viable path once the development of graphene-enhanced SSBs gains momentum. To start with, stable SSBs should be designed that could challenge conventional LIBs in terms of performance metrics. Graphene’s role should be further explored in enhancing the electrochemical and mechanical properties of the SSEs, electrodes, and interfaces. Since GBMs have different roles for each component of an SSB, careful consideration should be given to tailoring their properties that can benefit the relevant material’s performance. GO, so far, has been the preferred material in academic research for enhancing SPEs properties. Oxidizing or altering GO’s surface chemistry by introducing new functionalized groups might improve its interaction with polymer molecules but also create more mass-production regulatory uncertainties from the health and safety standpoint. For electrodes, controlling the size and chemistry of the graphene’s micro and nanostructures will be very important. The roles of rGOs and FLG will be significant, especially in the design of electrodes. Here rGO and FLG should offer high surface area providing greater cites for Li to intercalate. They should have high electronic conductivity to help reduce cell polarisation and facilitate redox activity. Lastly, graphene, as an interface mediator, should have the mechanical flexibility to buffer the Li anode expansion and ensure a robust interfacial contact with the SSE. They should also adhere to the uneven surfaces at the electrode|electrolyte solid-solid junctions to fill up micropores that will regulate Li^+^ ion flow and minimize the risk of Li dendrites.

Currently, the use of graphene in SSBs is still in an early stage. However, the near future is going to see an enormous demand for GBM-enhanced SSBs. To achieve the target performance, systematic research strategies should be developed, considering both performance and mass-production limitations. We believe that the pristine form of graphene is much more regulated, inexpensive, and commercially available, and therefore it is likely that more research will be performed on FLG and GNP in the near future, even though the properties of GO could be more suitable in certain cases. There will be a need for the development of more powerful in situ and operando characterization tools, which could provide insights into the interactions of GBMs with various components of an SSB. Further, computational studies as a supplement to experimental efforts may allow researchers to devise better strategies and design advanced solid-state energy storage systems.

## Figures and Tables

**Figure 1 nanomaterials-12-02310-f001:**
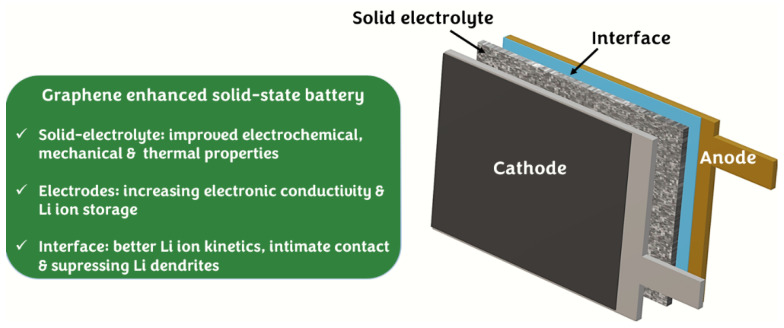
Schematic illustration of an SSB. Listed are the improvements imparted by graphene or GBMs relevant to electrodes, electrolyte, and interfaces.

**Figure 3 nanomaterials-12-02310-f003:**
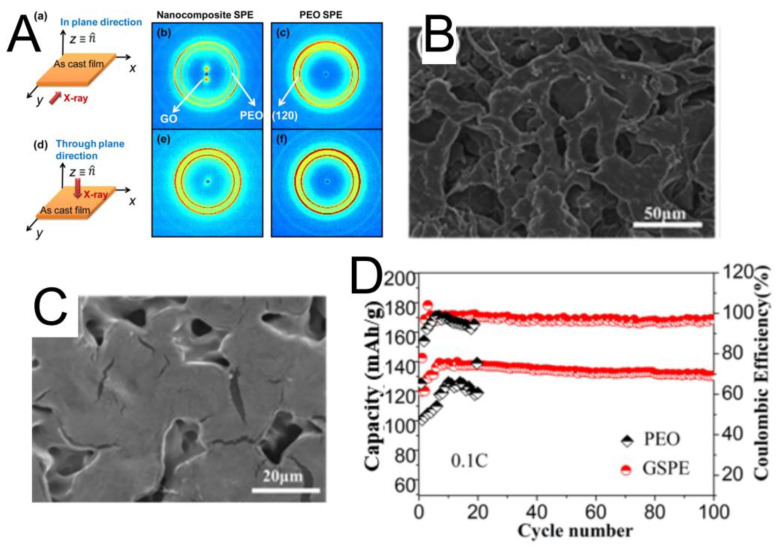
(**A**) 2D wide-angle XRD for GO-PEO and neat PEO samples (a–c) In-plane (d–f) Through-plane. Reprinted with permission [74]. Scanning electron microscopy (SEM) images of GO framework before and (**B**) After (**C**) PEO filling (**D**) Comparison of electrochemical performance for 3D graphene-PEO composite and only PEO. Reprinted with permission [76].

**Figure 4 nanomaterials-12-02310-f004:**
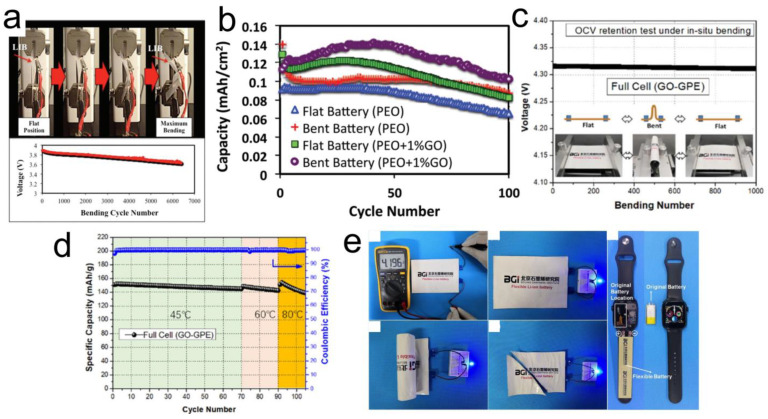
(**a**) Bending tests for GO-polymer composite electrolyte showing voltage response after more than 6000 bending cycles (**b**) Cycle life tests for SPEs with and without GO in flat and bend forms. Reprinted with permission [81]. (**c**) Graphite|GO-PTC SPE|LCO cell (**c**) Tested under mechanically deformed conditions (**d**) Cycle life tests at different temperatures, (**e**) Demonstrating safety of the battery and showcasing its application for an Apple watch. Reprinted with permission [82].

**Figure 5 nanomaterials-12-02310-f005:**
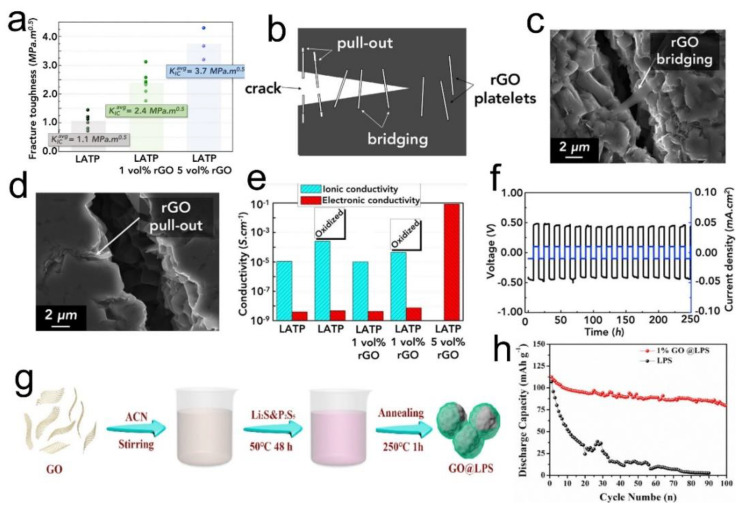
(**a**) Fracture toughness of monolithic and rGO-reinforced LATP, (**b**) Schematic illustration of the role of rGO in enhancing fracture toughness of LATP SSE, SEM images of LATP microstructure with (**c**) rGO bridging and (**d**) rGO pull out at the microcracks, (**e**) Ionic and electronic conductivity at various vol% of rGO, (**f**) Li strip/plat response of the SSE with 1 vol% rGO. Reprinted with permission [94]. (**g**) Schematic representation of the synthesis process of GO@LPS materials, (**h**) Comparison of cell performances of LPS coated with 1 wt% GO and without GO. Reprinted with permission [95].

**Figure 7 nanomaterials-12-02310-f007:**
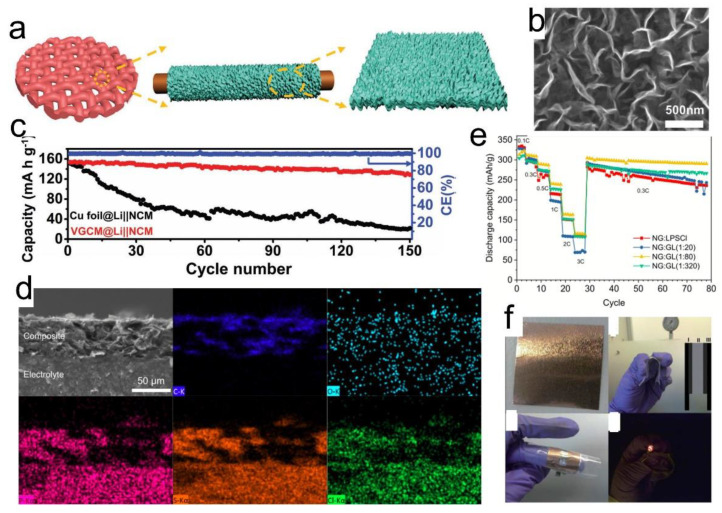
(**a**) Schematic illustration of the synthetic procedure of the VGCM; (**b**) Magnified SEM image of the vertical graphene nanoarray; (**c**) Electrochemical cycling performance of cells based on Cu foil @Li|NCM and vertical graphene on Cu mesh @Li|NCM. Reprinted with permission [110]. The 2D rGO- Li_6_PS_5_Cl -graphite composite electrode (**d**) Cross-sectional SEM and EDS images (**e**) Rate performance. Reprinted with permission [111]. (**f**) Photographs of monolayer graphene grown on Cu foil, a flexible graphene battery in the bent state, the battery powering a LED. Reprinted with permission [112].

**Figure 8 nanomaterials-12-02310-f008:**
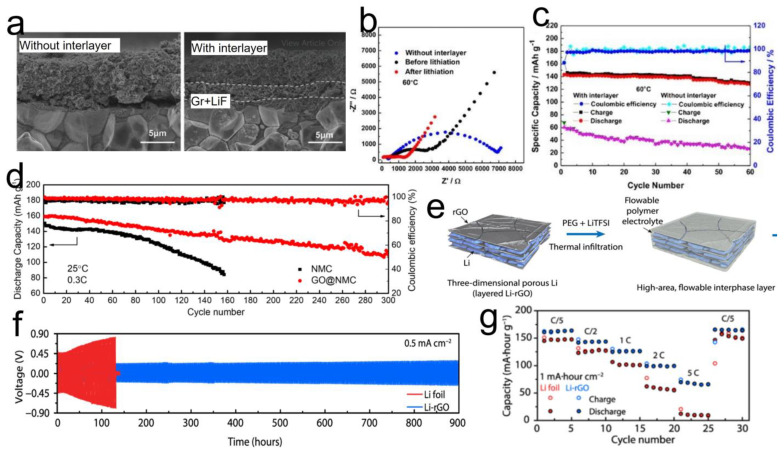
(**a**) SEM images of LLZTO|LFP interface with graphene-LiF as an interlayer (shown by dotted lines); (**b**) EIS spectra for samples with and without Gr-LiF interlayers; (**c**) Cyclic life tests of cells with and without graphene-LiF interlayers; Reprinted with permission [128]. (**d**) Electrochemical performance comparisons for NMC and GO modified NMC cathodes in SSB; Reprinted with permission [106]. (**e**) Schematic representation of 3D rGO-Li electrode; (**f**) Li strip/plat voltage profiles for symmetric cells based on Li and Li-rGO electrodes; (**g**) Rate-tests to compare performance of batteries based on Li and Li-rGO electrodes when paired with LFP cathodes. Reprinted with permission [132].

**Table 1 nanomaterials-12-02310-t001:** An overview of graphene and related materials relevant to the electrolyte, electrode, and interface of a solid-state batteries.

SSB Component	GBM Type and Role	Enhancements	Ref.
Electrolyte			
PEO	GO as randomly oriented filler	The Li|LFP cell delivered an initial discharge capacity of 142 mAh g^–1^ at 0.5 C and 91% capacity retention after 100 cycles	[59]
PAN	GO as randomly oriented filler	LFP cells with GO-PAN electrolytes delivered 166 mAh g^−1^ at 0.2 C that was higher than cells without GO (136 mAh g^−1^)	[62]
PEO	3D GO aerogel as dimensional filler	Ionic conductivity of 4 × 10^−4^ S cm^−1^. Li symmetrical cells cycles for >600 h at 0.1 mA cm^−2^; LFP cell @ 50 °C delivered ~130 mAh g^−1^ after 100 cycles; Cells without GO electrolytes short-circuited after few cycles.	[76]
PTC	GO-PTC composites as flexible electrolyte	Cell OCV maintained for >1000 cycles under bend and flat conditions; LCO cell delivered ~120 mAh g^−1^ capacity at 25 °C and >150 mAh g^−1^ at 45 °C; demonstration under flat, rolled, and cut conditions	[82]
LATP	Nano-rGO reinforces ceramic electrolyte	Solid electrolyte mean toughness increased from 1.1 MPa.m^0.5^ for 1 vol% to 2.4 MPa.m^0.5^, for 5 vol%; overpotential stayed constant over 250 h of cycling	[94]
Electrode			
S-rGO	S-rGO composite electrode	rGO improved e^-^ conductivity; buffered electrode volume change; traps polysulfides; delivered ~1100 mAh g^−1^ at 1.0 C; retented of 830 mAh g^−1^ at 1 C after 750 cycles	[100]
Li-rGO-Cu(VGCM)	Vertical graphene grown on Cu-based	High and stable coulombic efficiency for 250 cycles at 2 mA cm^−2^. Bare Cu foil showed a rapid decrease in within 50 cycles; Li symmetrical cell had a small overpotential of ~35 mV after 500 h cycling; stability VGCM@Li||NCM showed capacity retention of 83.79% after 150 cycles	[110]
rGO- Li_6_PS_5_Cl-graphite	2D rGO acts as tempelate for Li_6_PS_5_Cl	2D Li_6_PS_5_Cl particles with a high-aspect ratio developed using rGO as a tempelate; improved rate performance and capacity retention was achieved for the composite electrodes	[111]
Interface			
LFP-Gr-LiF-LLZO	Graphene-LiF composite as interlayer	At 60 °C the cells with interlayer showed resistance of 3502 Ω cm^2^ (before lithiation), 1538 Ω cm^2^ (after lithiation); without interlayer showed 7829 Ω cm^2^; the cell tested at 60 °C delivered 1^st^ discharge capacity of 143 mAh g ^−1^ and a capacity retention rate of 90% after 60 cycles	[128]
Li-rGO	rGO as interlayer	Wide potential window (~4.8 V), ionic conductivity (~2 × 10^–4^ S cm^–1^) and a high ionic transference (0.9); For NMC|Li cell initial capacity: 160 mAh g^−1^ at 0.5 C; some capacity decay, (100 mAh g^−1^) after 200 cycles	[131]
Li-rGO	As interlayer between Li and SPE	>900 h of stable Li cycling; bare Li-based electrodes short-circuited before 150 h; significantly improved rate performance: at 2 C ~100 mAh g^−1^ for Li-rGO vs. ~50 mAh g^−1^ for bare Li foil	[132]

## Data Availability

Not applicable.

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
