# Peer review of "Graphene in Solid-State Batteries: An Overview"

_nanomaterials, 2022, doi:10.3390/nano12132310_

Round 1

Reviewer 1 Report

The manuscript “Graphene in solid-state batteries: an overview” by S. A. Pervez et al. presents an overview of the role of graphene-based materials in improving the performance of solid-state batteries (SSBs) with a focus on the recent progress in this field. The authors describe in some detail the components of SSBs and highlight several approaches for the improvement of the batteries' stability and performance. Also, they analyze the properties of related materials relevant to the electrolyte, electrodes, and interfaces of SSBs.

I think that this work presents a valuable contribution and may be suitable for publication. The manuscript is well written and presents the data in a convincing manner. I suggest to the authors address two points:

1. The size of all the panels in all the figures should be increased. In the present shape, it is difficult to distinguish the details.

2. Many references, like 9, 13, 31, 42, and many others, lack the publication name.

Reviewer 2 Report

This review paper summarizes the application of graphene (oxides) in solid state batteries, in terms of electrolyte, anode, cathode and electrode/electrolyte interfaces. At beginning, a description of the graphene fabrication methods is provided, which shall be commended for non-expert readers. The functions and challenges of graphene in SSB systems are systematically discussed with well-organized logic and language. Therefore, I would like to recommend its publication on Nanomaterials after addressing my below minor concerns:

1. Small amounts of GO are widely used to optimize the ionic conductivity, mechanical flexibility and interfacial stability in various solid state electrolytes. The percolation, size and morphology of GO are considered important information to readers following this review to design their GO-modified SSEs. Thus, I would like to recommend the authors to add more discussion or literature survey on these aspects.

2. In Page 9, it seems 1wt% GO additive can significantly regulate the flexibility of thin film batteries. How does such marginal additive significantly tune the mechanical property of SSE? Please check and clarify.

3. In regard to the section “graphene in cathodes”, I cannot find examples for graphene/lithium transition metal oxide cathodes (i.e., LiFePO4, NMC), which are widely investigated in SSBs. In addition, the paragraph in Page 13 discussion rGO-S composite is suggested to move above Li-S battery sections.

4. Some typos: Page 4: “graphene or graphene-based materials (GBMs) exhibits novel” Page 6: “used to control the crystal growth”

5. Some related literature are suggested to be added: Journal of Materials Chemistry 21.26 (2011): 9762-9767; Materials Today Chemistry 25 (2022): 100967; Microstructures 2.10 (2022): 1.

Reviewer 3 Report

The review article on " Graphene in solid state batteries: an overview " provides the readers with up-to-date information on the aspects that have not been covered previously. Below are some of the comments that authors should look into improving the manuscript.

  1. The authors should add a table including GO and its modifications and performances from the references included in the review.
  2. The authors should provide more information about possible future directions.
  3.  The quality of figures should be enhanced as some of the wordings are unclear.
  4. It would be better to redraw the Schematic of a SSB in Figure.1 showcasing all the key highlights of the article
  5. There are several mistakes, authors should check for spellings, symbols that are wrongly placed, space between words etc.
